# Diagnostic Accuracy of Dual-Energy CT Material Decomposition Technique for Assessing Bone Status Compared with Quantitative Computed Tomography

**DOI:** 10.3390/diagnostics13101751

**Published:** 2023-05-16

**Authors:** Xu Wang, Beibei Li, Xiaoyu Tong, Yong Fan, Shigeng Wang, Yijun Liu, Xin Fang, Lei Liu

**Affiliations:** Department of Radiology, First Affiliated Hospital of Dalian Medical University, Dalian 116014, China; wx6760107@163.com (X.W.); 18895663700@163.com (B.L.); tongxy1226@163.com (X.T.); fyfan_yong@163.com (Y.F.); wangshigeng9855@163.com (S.W.); yijunliu1965@126.com (Y.L.); fx25632331@163.com (X.F.)

**Keywords:** bone density, osteoporosis, quantitative computed tomography (QCT), dual-energy computed tomography (DECT), material decomposition

## Abstract

Purpose: The purpose of this study was to evaluate the diagnostic accuracy when using various base material pairs (BMPs) in dual-energy computed tomography (DECT), and to establish corresponding diagnostic standards for assessing bone status through comparison with quantitative computed tomography (QCT). Methods: This prospective study enrolled a total of 469 patients who underwent both non-enhanced chest CT scans under conventional kVp and abdominal DECT. The bone densities of hydroxyapatite (water), hydroxyapatite (fat), hydroxyapatite (blood), calcium (water), and calcium (fat) (D_HAP (water)_, D_HAP (fat)_, D_HAP (blood)_, D_Ca (water)_, and D_Ca (fat)_) in the trabecular bone of vertebral bodies (T11–L1) were measured, along with bone mineral density (BMD) via QCT. Intraclass correlation coefficient (ICC) analysis was used to assess the agreement of the measurements. Spearman’s correlation test was performed to analyze the relationship between the DECT- and QCT-derived BMD. Receiver operator characteristic (ROC) curves were generated to determine the optimal diagnostic thresholds of various BMPs for diagnosing osteopenia and osteoporosis. Results: A total of 1371 vertebral bodies were measured, and QCT identified 393 with osteoporosis and 442 with osteopenia. Strong correlations were observed between D_HAP (water)_, D_HAP (fat)_, D_HAP (blood)_, D_Ca (water)_, and D_Ca (fat)_ and the QCT-derived BMD. D_HAP (water)_ showed the best predictive capability for osteopenia and osteoporosis. The area under the ROC curve, sensitivity, and specificity for identifying osteopenia were 0.956, 86.88%, and 88.91% with D_HAP (water)_ ≤ 107.4 mg/cm^3^, respectively. The corresponding values for identifying osteoporosis were 0.999, 99.24%, and 99.53% with D_HAP (water)_ ≤ 89.62 mg/cm^3^, respectively. Conclusions: Bone density measurement using various BMPs in DECT enables the quantification of vertebral BMD and the diagnosis of osteoporosis, with D_HAP (water)_ having the highest diagnostic accuracy.

## 1. Introduction

Osteoporosis is a metabolic bone disease characterized by reduced bone mineral density, bone mass, and bone microarchitecture [1,2]. With the growth of the global population and the aging problem, the incidence of osteoporosis among the elderly population has become a global public health issue. Fragility fractures caused by osteoporosis, especially vertebral fractures, are associated with high mortality and disability rates [3]. However, the majority of osteoporosis patients are asymptomatic at an early stage, so early diagnosis and interventions through medical imaging examinations are critical to prevent the occurrence of complications [4].

The most important indicator for the diagnosis of osteoporosis is the bone mineral density (BMD), decreases in which are directly related to osteoporosis [5]. Although the current WHO-recommended method for diagnosing osteoporosis is dual-energy X-ray absorptiometry (DXA) [6], DXA is unable to accurately reflect changes in spongy bone and is susceptible to calcification of the ligaments and abdominal aorta, leading to false negative results [7]. Quantitative computed tomography (QCT) can obtain the true volumetric bone density independent of bone size, morphology, vertebral degenerative changes, and vascular calcification, and is characterized by high sensitivity and accuracy [8,9]. However, QCT requires dedicated CT scans and special calibration and analysis software.

In recent years, dual-energy computed tomography (DECT) has been widely used in clinical routine imaging examination, with increasing use in clinical practice, and its role is becoming more and more important. DECT can achieve not only the function of conventional CT but also multi-parameter imaging, including energy spectrum curves, material decomposition, monochromatic images, and the effective atomic number, among other technical advantages [10]. It also has unique advantages for the musculoskeletal system, such as detecting bone marrow edema in traumatic and non-traumatic environments, diagnosing vertebral compression fractures, and identifying bone marrow tumors [11]. Material decomposition refers to the fact that X-ray attenuation images scanned with high and low voltages can be expressed as density maps of two base substances, and each voxel in the separated material density image reflects the corresponding material density information. Studies have shown that the absorption of any type of tissue can be determined by the proportion of its base material, so the value of the appropriate base material pairs (BMPs) can represent the actual material content in the tissue [12]. Measurement of bone mineral density with BMPs has become a new method for bone mineral density research, which has broad prospects in the quantification of bone mineral density. DECT is expected to help in carrying out one-stop examination, which can not only diagnose vertebral fractures, but also obtain vertebral bone mineral density information.

Previous studies on BMD measurement in DECT have explored the feasibility of using a particular base material pair to respond to changes in BMD. Li et al. [13] compared the accuracy of DECT and QCT for BMD measurement using the European spine phantom (ESP). The results showed that they could maintain a high accuracy in BMD measurement, with the condition of a smaller deviation in DECT when using HAP (water) as the base material pair. Yue et al. [14] reported that Ca (water) could reflect the age-related changes in the lumbar spine of adult women and the correlation with BMD; hence, the calcium (water) density could be used for BMD evaluation. To date, few studies have conducted comprehensive cross-sectional comparisons of multiple different BMPs. The choice of BMP requires further exploration due to the complex composition of vertebrae. Because of the different measuring methods, the diagnostic criteria of QCT for osteoporosis cannot be directly applied to the BMD values given by DECT, and vice versa. There is still no universal agreement on the threshold value in DECT for diagnosing osteoporosis and osteopenia. Our study performed base material pairing of the major components of the vertebrae and analyzed six base material pairs: HAP (water), HAP (fat), HAP (blood), Ca (water), Ca (fat), and fat (HAP). While using QCT as a reference standard to establish the threshold values for each base material pair to determine the corresponding bone status, our study aimed at evaluating the efficacy of predicting osteoporosis and normal bone status by using BMD measurements with different BMPs in DECT.

## 2. Materials and Methods

### 2.1. Population and Study Design 

This study was approved by the ethics committee of the First Affiliated Hospital of Dalian Medical University (PJ-KS-KY-2022-43). Informed consent was obtained from all subjects involved in the study. Continuous patients who underwent both a non-enhanced chest CT under conventional kVp and abdominal DECT from March 2022 to July 2022 were collected. The time interval between the chest and abdominal CT examinations was within a month to make sure that the bone density of the vertebrae was stable over a short period of time. There was anatomical overlap between the chest and abdominal CT scans that were used to calculate the QCT-based BMD (from the chest CT with dedicated QCT software) and DECT-derived BMD (density measurement from the abdominal DECT) of three vertebrae (T11–L1) for direct comparison. The subjects’ demographic characteristics (age, sex) before scanning were recorded.

Exclusion criteria: history of lumbar spine surgery (implants, hardware, or other foreign material); malignancy; abnormal spine morphology (lumbar compression fracture and scoliosis) or severe degenerative changes; diseases affecting bone density (rheumatic diseases, endocrine diseases, hematologic disorder, etc.) [9,15]. All images were read by experienced radiologists with 10 years of experience in spine CT. Information on other factors thought to affect bone density was obtained through a review of clinical case records. A total of 469 patients were finally enrolled in the study project. The mean time interval between the two examinations was 20 days (range = 5–25 days). A flow chart detailing the stepwise exclusion of the subjects and vertebral bodies is presented in Figure 1.

### 2.2. CT Imaging Protocol

All CT scans were performed using a clinical Revolution CT scanner (GE Healthcare, Milwaukee, WI, USA). The scanning parameters for the conventional chest CT were as follows: tube voltage, 120 kVp; 3D smart mA; range 100–600 mA; scanning range from the lung apex to the lower level of the L1 vertebral body. The scanning parameters for the abdominal DECT were as follows: fast-switching kVp (80 kVp/140 kVp) within 0.5 ms; gemstone spectral imaging (GSI) assist for modulating mA; scanning range from above the diaphragm to the inferior margin of the liver or pubic symphysis. The same noise index of 11 was set for the chest and abdominal scans, with the application of 40% pre-adaptive statistical iterative reconstruction-volume (ASIR-V) in order to reduce the radiation dose. The other parameters were kept the same: detector width, 80 mm; rotational speed, 0.5 s/r; pitch, 0.992:1; matrix, 512 × 512; slice thickness, 5 mm.

### 2.3. Image Reconstruction and Quantitative Data Measurement

The raw data of the chest examination were reconstructed using a standard algorithm with a reconstruction slice thickness of 1.25 mm and DFOV of 50 cm. For the abdominal DECT scan data, both the 70 keV monochromatic images and material decomposition images at the 1.25 mm slice thickness using the BMPs of hydroxyapatite–water, hydroxyapatite–fat, hydroxyapatite–blood, calcium–water, and calcium–fat were reconstructed.

QCT-based BMDs and material densities in the material decomposition images of various BMPs in DECT were measured at the T11, T12, and L1 vertebrae, which were superimposed on the chest and abdominal scans of the same patient. 

The Mindways Model 4 QCT Pro V6.1 Bone Densitometry System (Mindways QCT Pro; Mindways Software, Inc., Austin, TX, USA) was used for BMD measurements to characterize patients in terms of bone status. According to the terms of use, the QCT system was calibrated at regular intervals of one week by scanning the phantom under chest CT parameters. The conventional CT images were transferred to a QCT workstation. Trabecular bone with a uniform, normal density at the middle level of the vertebrae was selected as the measurement area. A volume of interest (VOI) with a height of 9 mm and cross-sectional area of ≥100 mm^2^ was set and then adjusted in the axial, coronal, and sagittal planes. The range of the VOI was within the edge of the bone cortex encompassing as much cancellous bone as possible, with an effort to avoid bone islands and the posterior vertebral plexus (Figure 2). BMD was obtained using the processing and analysis software on the QCT workstation. The QCT BMD values (unit: mg/cm^3^) were recorded for each vertebra. Volumetric trabecular (vBMD) measurements from 80 to 120 mg/cm^3^ are defined as osteopenia, and vBMD measurements below 80 mg/cm^3^ are defined as osteoporosis, based on the guidelines. This criterion has been validated in the Chinese population [16].

The 1.25 mm monochromatic images and material decomposition images with various BMPs were transferred to an AW4.7 workstation (Advantage Windows 4.7, GE Healthcare, Milwaukee, WI, USA). The density values of HAP were measured in the HAP (water), HAP (fat), and HAP (blood) material decomposition images, while the density values of calcium were measured in the Ca (water) and Ca (fat) images. In addition, the density values of fat were measured in the fat (HAP) images. The absorption of any type of tissue can be determined by the proportion of its base material, so the value of the appropriate base material pairs (BMPs) can represent the actual material content in the tissue. A region of interest (ROI) with an area of about 100 mm^2^ was placed in the uniform, normal-density area of spongy bone at the middle level of T11, T12, and L1, with a distance of over 5 mm to the edge of the vertebrae, avoiding the vertebral venous plexus and bone islands [17] (Figure 3). Three consecutive levels from above to below the middle level of the vertebrae were measured to obtain the average value, with the unit of mg/cm^3^, to represent the vertebral BMD.

The QCT and DECT data were evaluated in random order by two radiologists with more than 10 years of experience in musculoskeletal radiology, who worked independently and were blinded to each other’s results. The data were measured twice by each radiologist.

### 2.4. Statistical Analysis

The statistical analyses were performed using SPSS software, version 24.0 (SPSS Inc., IBM, Armonk, NY, USA). Statistical analysis was performed independently by a physician with 3 years of experience in medical statistics. The numeric variables are represented by the mean ± standard deviation. A single vertebra was used as the study subject. The agreement of the data measurements between the two physicians was assessed using the intraclass correlation coefficient (ICC) and 95% confidence interval (CI). When the agreement between the two observers was acceptable (ICC value > 0.75), the group with a smaller standard deviation was used for the subsequent statistical analysis of the data. Spearman’s correlation coefficient was used to evaluate the association between the density value measurement of various BMPs and the QCT-based BMD. Using the diagnostic results of QCT on vertebra mass as a gold standard, the vertebrae were divided into three groups: normal bone, osteopenia, and osteoporosis. The Kruskal–Wallis test was used to compare the statistical differences in the three groups of vertebrae between the QCT-based BMD and DECT-based density measurements. Medcalc version 19.0 (MedCalc, Ltd., Ostend, Belgium) statistical software was used to generate the receiver operating characteristic (ROC) curves to evaluate the diagnostic accuracy of using the density values in each base material pair in the vertebrae for evaluating osteoporosis, osteopenia, and normal bone determined by QCT. Threshold values for the density value were established for obtaining the highest Youden index, and the sensitivities and specificities are reported. A two-sided *p* value of <0.05 was regarded as statistically significant.

## 3. Results

### 3.1. Populations

A total of 1371 vertebral bodies of 469 participants were measured after exclusion screening. Among them, 54.6% of the vertebral bodies were from male patients, and 45.4% were from female patients. T11, T12, and L1 represented 22.44%, 29.39%, and 34.21% of the vertebral bodies in osteoporosis, respectively, and 29.85%, 31.80%, and 35.09% of the vertebral bodies in osteopenia, respectively (Table 1).

### 3.2. The Difference among Base Material Pairs and Correlation between DECT- and QCT-Derived BMD

The agreement of the data measurements was evaluated by comparing the QCT-derived BMD and the density values of each base material pair measured twice by both physicians. The agreement of the above-mentioned statistics between the two radiologists was acceptable (ICC values = 0.976–0.990, both >0.75) (Table 2).

Both inter-group comparisons demonstrated statistically significant differences in the densities of the base material pairs in the vertebrae (*p* < 0.05, Table 3). Furthermore, there were statistically significant differences in the density values between the osteoporosis, osteopenia, and normal bone mass groups (*p* < 0.05).

Figure 4 shows the correlation between the QCT-derived BMD values and the values of each base material pair. Significant positive correlations were demonstrated between D_HAP (water)_, D_HAP (fat)_, D_HAP (blood)_, D_Ca (water)_, and D_Ca (Fa)_ and the QCT-derived BMD (r = 0.922–0.942, all *p* < 0.001). On the contrary, the correlation between D_Fat (HAP)_ and the QCT-derived BMD was weaker (r = 0.156, *p* < 0.001).

### 3.3. Diagnostic Effectiveness Evaluation Using Base Material Pairs Derived from DECT

The area under the curve (AUC) for using D_HAP (Water)_, D_HAP (fat)_, D_HAP (blood)_, D_Ca (water)_, D_Ca (fat)_, and D_Fat (HAP)_ for diagnosing osteopenia was 0.953, 0.930, 0.934, 0.930, 0.932, and 0.556, respectively, with the AUC for using D_HAP (Water)_ being the largest (Figure 5a). Using D_HAP (Water)_ ≤107.4 (mg/cm^3^) as the threshold, the sensitivity and specificity were 86.88% and 88.91%, respectively. The differences in the AUC between D_HAP (Water)_ and the rest of the base material pairs were statistically significant (Z = 3.401~21.823, *p* < 0.05). Table 4 demonstrates the ROC results of using various BMPs for diagnosing osteopenia.

The AUC for using D_HAP (water)_, D_HAP (fat)_, D_HAP (blood)_, D_Ca (water)_, D_Ca (fat)_, and D_Fat (HAP)_ for diagnosing osteoporosis was 0.999, 0.996, 0.997, 0.997, 0.997, and 0.594, respectively, with the AUC for using D_HAP (Water)_ being the largest (Figure 5b). Using D_HAP (water)_ ≤ 89.62 (mg/cm^3^) as the threshold, the sensitivity and specificity for recognizing osteoporosis were 99.24% and 99.53%, respectively, with the AUC of the ROC curve being 0.999. The differences in the AUC between D_HAP (water)_ and the rest of the base material pairs were statistically significant (Z = 3.369~19.728, *p* < 0.05). Table 5 demonstrates the ROC results of using various BMPs for diagnosing osteoporosis.

## 4. Discussion

The early diagnosis of osteoporosis has attracted considerable attention [4]. Our study shows that the energy spectral CT material decomposition technique can accurately quantify vertebral bone mineral density, which is a new method for BMD measurement, and that HAP (water)-, HAP (fat)-, HAP (blood)-, Ca (water)-, and Ca (fat)-based material pairs can be used for the assessment of bone quality status, with HAP (water) having the best ability to predict osteoporosis. This study also provides a preliminary diagnostic threshold reference for the use of BMPs in the diagnosis of osteoporosis and osteopenia.

In clinical studies, the radiation dose from a CT examination is non-negligible. Previous studies always required patients to complete two separate BMD examinations in succession to achieve the purpose of making comparisons between different modalities for BMD quantification. In contrast, our study innovatively used the data from the overlapping portion between the chest and abdominal CT examinations to obtain BMD values measured via QCT and DECT, with no additional radiation dose for the patients.

According to the consensus of QCT application, it is recommended to collect the mean BMD values from two intact vertebral bodies in the range of T12 to L3 to demonstrate the patient’s bone status [16]. However, the purpose of our study was to compare the differences in BMD measurements between the QCT-based BMD and DECT-based density measurements. Moreover, during the QCT BMD measurements, it was found that determining bone status using mean BMD values equal to or greater than two vertebrae would lead to an underestimation of the bone status in patients with a low bone mass, which is the reason why a single vertebra was the analysis object in our study. The major fracture location in osteoporotic patients is the junction of the thoracolumbar spine, which is likely to be associated with higher mechanical stress [18]. The proportions of T11, T12, and L1 in both osteoporosis and osteopenia were calculated via QCT measurements, after grouping according to the QCT diagnostic criteria. The findings suggest that there is an evident increase from T11 to L1, which may be related to the structural characteristics of the human vertebrae. A good case in point for this is that L1 is the first vertebra without rib connection, and chest and abdominal CT examinations generally include the L1 vertebra, which was previously considered the best target for screening for opportunistic osteoporosis in routine CT examinations, and is easy to identify and less prone to degenerative changes than other vertebrae [19,20].

The major components of the vertebrae include bone minerals (hydroxyapatite/calcium), water, red marrow, yellow marrow (mainly fat), and collagen [21]. Leon D. et al. used an algorithm based on the biophysical model of material decomposition based on DECT, which explained the five main substances of trabecular bone and could accurately obtain the volume bone density of vertebral bodies, which is of great value in preventing osteoporotic fractures [22]. Pairing several of these major component pairs revealed that rapid kVp-switching DECT can accurately quantify vertebral bone density. D_HAP (water)_, D_HAP (fat)_, D_HAP (blood)_, D_Ca (water)_, and D_Ca (fat)_ all showed significant positive correlations with the QCT-derived BMD, which can be used to diagnose osteoporosis. Furthermore, the area under the ROC curve (AUC) was close to 1, which is consistent with the disease diagnostic test, and the D_HAP (water)_-specific density showed the best predictive ability. In terms of correlation, Zhou et al. indicated that the correlation between density measurements in material decomposition images in DECT and BMD values measured via QCT was high, which is consistent with the results of the present study [23]. Wichmann et al. showed a lack of correlation between BMD values measured via DECT and BMD values measured via DXA in 40 subjects with 160 vertebrae in total [24]. The inconsistency between their findings is probably due to the small sample size and the measurement of the area density via DXA. Our study population consisted of 436 adults of various ages with a total of 1372 vertebrae, which is a larger sample size compared with the above-mentioned research, meaning that more representative results may have been obtained. When using D_HAP (blood)_ to diagnose osteoporosis, we obtained a sensitivity of 99.24% and a specificity of 88.35%. The threshold value of D_HAP (blood)_ ≤ 86.91 was closest to the threshold value of 80 for diagnosing osteoporosis via the QCT-derived BMD. On the other hand, the sensitivity of using D_HAP (water)_ at a threshold of ≤89.62 was comparable to that of HAP (blood), while the specificity of 99.53% for D_HAP (water)_ was much higher. We believe the reason is that the proportion of water in the vertebrae is greater compared with blood [21]. Thus, D_HAP (water)_ can reflect the composition of vertebrae more adequately, meaning that it can be used to diagnose osteoporosis more accurately, which corresponds to the finding of the highest correlation between D_HAP (water)_ and BMD values in our study. Zhou et al. demonstrated that D_Ca (fat)_ and D_HAP (fat)_ have similar, optimal abilities to predict osteoporosis [23], which differs from the choice of the best base material pair in our study, probably because of the simultaneous QCT technique they used for the analysis of BMD.

There is an essential association between bone and fat, which means if the percentage of adipose tissue in the vertebral bone marrow increases, it will lead to a decrease in osteoblast production and a decrease in bone strength, therefore ultimately resulting in osteoporosis [25,26]. Hence, measuring the changes in fat content is of great importance in diagnosing osteoporosis. Zhao et al. used the mDixon quantification method to show that the fat fraction and T* values in the vertebrae had significant negative correlations with BMD [27]. In our study, fat (HAP) was specifically analyzed in an attempt to illustrate the trend of BMD through the changes in lipid content, but the results showed that fat (HAP) was not ideal for assessing the bone marrow fat content within the vertebrae. Although the difference in D_Fat (HAP)_ was statistically significant under different bone statuses, the AUC for diagnosing osteoporosis was 0.594, with low sensitivity and specificity, presumably because DECT is an indirect measurement of fat content through BMPs. Since MRI has high soft tissue resolution, it is more sensitive than DECT in showing changes in the vertebral fat content [28].

The advantage of our study is that dual-energy CT substance separation technology was used to evaluate the correlation between BMPs and the QCT-based BMD, and based on QCT, the diagnostic threshold of different base substances for the diagnosis of osteoporosis and osteopenia was preliminarily obtained, which may be implemented in the clinical routine. Although there have been studies using base pairs to assess vertebral bone status, to date, comprehensive analyses of BMPs have been insufficient, and the use of BMPs to assess bone states remains an issue of high scientific and clinical interest.

There are several limitations in this study that require further improvements. Firstly, the composition of trabecular bone is complex, and current BMD measurements can only provide rough estimates. To pursue improvements in the measurement accuracy, however, densitometry via chemical analysis is not applicable. Secondly, this study only analyzed T11, T12, and L1 to derive diagnostic threshold values for each base material pair corresponding to QCT BMD as a reference. Therefore, the inclusion of vertebrae should be expanded. In addition, the results of our study need to be further validated in a multicenter study with a larger sample population.

In conclusion, bone density measurement using various BMPs in DECT enables the accurate quantification of the vertebral BMD and high diagnostic efficacy for osteoporosis. The use of D_HAP (Water)_ measurement for the hydroxyapatite–water base material pair provides the highest diagnostic accuracy. 

## Figures and Tables

**Figure 1 diagnostics-13-01751-f001:**
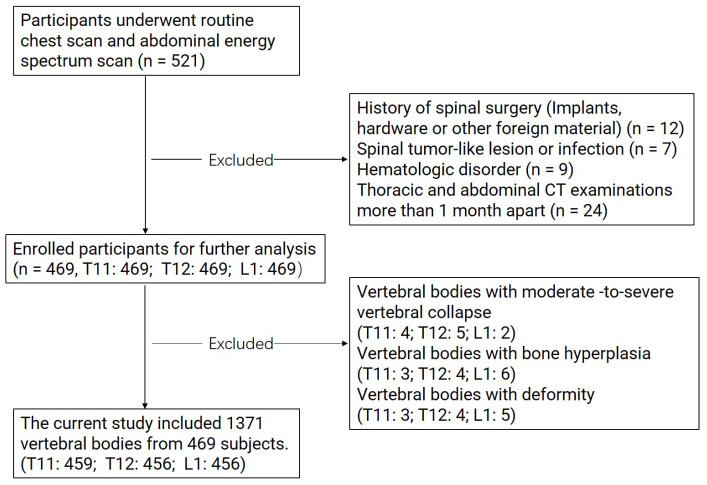
Flow chart illustrating the stepwise exclusion of subjects and vertebral bodies.

**Figure 2 diagnostics-13-01751-f002:**
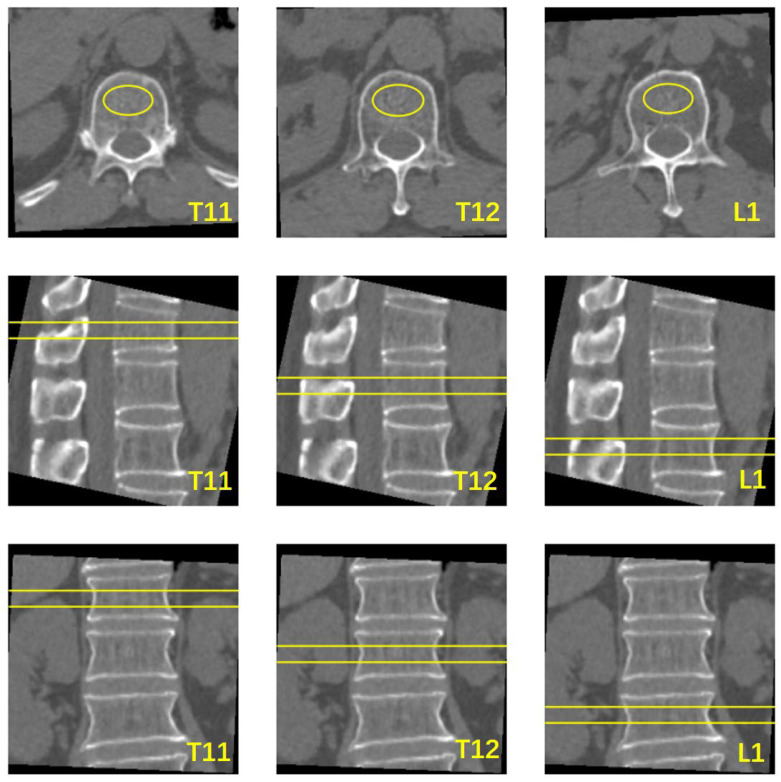
Schematic diagram of vertebral BMD measured using QCT software. The yellow oval area in the axial section is the cross-sectional area of VOI, the area of the yellow box line in the sagittal and coronal planes is the area of the vertebral body measured by VOI.

**Figure 3 diagnostics-13-01751-f003:**
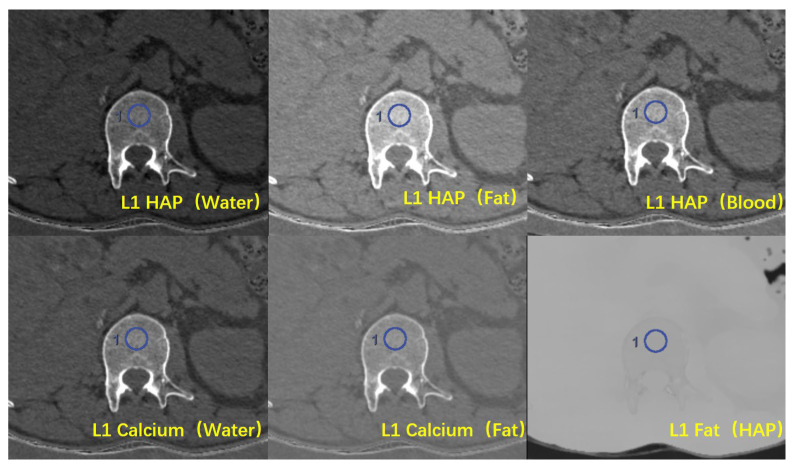
Six types of material decomposition (MD) images. Blue circles illustrate the ROI location in the vertebral body.

**Figure 4 diagnostics-13-01751-f004:**
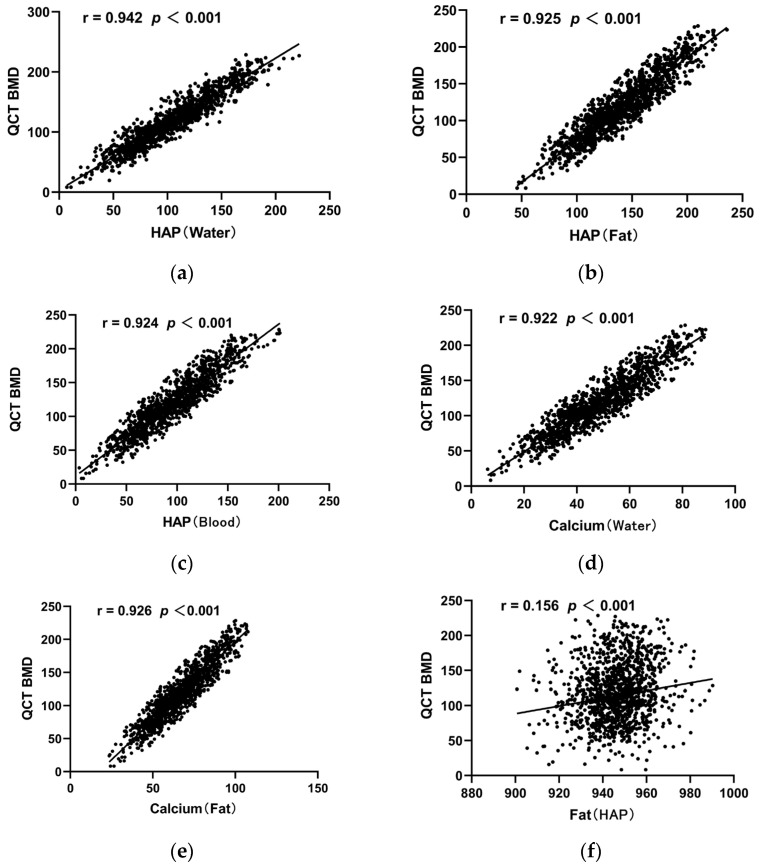
Scatter diagram of correlation coefficients between the QCT-derived BMD of vertebral bodies and each base material pair. (**a**) Scatter diagram of correlation coefficients between QCT BMD of vertebral body and HAP (water); (**b**) Scatter diagram of correlation coefficients between QCT BMD of vertebral body and HAP (Fat); (**c**) Scatter diagram of correlation coefficients between QCT BMD of vertebral body and HAP (Blood); (**d**) Scatter diagram of correlation coefficients between QCT BMD of vertebral body and Calcium (water); (**e**) Scatter diagram of correlation coefficients between QCT BMD of vertebral body and Calcium (Fat); (**f**) Scatter diagram of correlation coefficients between QCT BMD of vertebral body and Fat (HAP).

**Figure 5 diagnostics-13-01751-f005:**
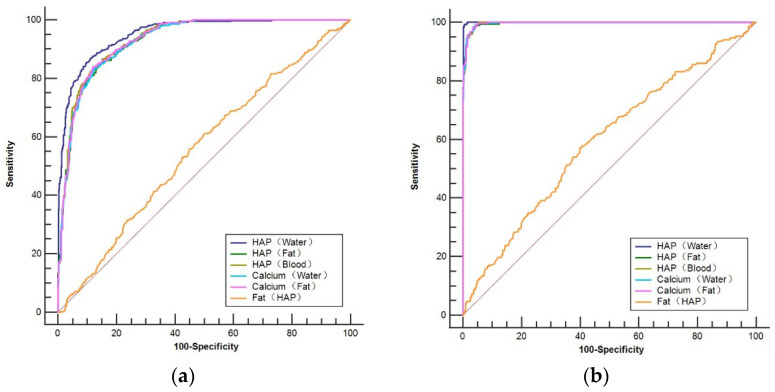
ROC curve for predicting osteopenia (**a**) and osteoporosis (**b**) based on vertebral base material pair density values compared with the QCT-derived BMD.

**Table 1 diagnostics-13-01751-t001:** Characteristics of participants.

Participants (*n* = 469)	Sex	Male (*n* = 256)
		Female (*n* = 213)
	Average Age (y)	63.24 (22~90)
Vertebral bodies (*n* = 1371)	Osteoporosis (*n* = 393)	T11:103; T12:134; L1:156
	Osteopenia (*n* = 442)	T11:137; T12:145; L1:160
	Normal (*n* = 536)	T11:219; T12:177; L1:140

**Table 2 diagnostics-13-01751-t002:** Consistency of data measurements between two radiologists.

	ICC	ICC 95% ICC	*F*	*p*
QCT BMD (mg/cm^3^)	0.990	(0.984, 0.992)	798.488	0.001
HAP (water) (mg/cm^3^)	0.977	(0.952, 0.988)	154.252	0.001
HAP (fat) (mg/cm3)	0.981	(0.947, 0.991)	209.703	0.001
HAP (blood) (mg/cm^3^)	0.978	(0.936, 0.989)	189.071	0.001
Ca (water) (mg/cm^3^)	0.976	(0.954, 0.991)	206.772	0.001
Ca (fat) (mg/cm^3^)	0.985	(0.957, 0.990)	172.767	0.001
Fat (HAP) (mg/cm^3^)	0.980	(0.973, 0.985)	60.168	0.001

**Table 3 diagnostics-13-01751-t003:** Comparison of density values of various base material pairs in vertebral bodies under different bone states.

Bone States	HAP (Water)	HAP (Fat)	HAP (Blood)	Ca (Water)	Ca (Fat)	Fat (HAP)
	(mg/cm^3^)	(mg/cm^3^)	(mg/cm^3^)	(mg/cm^3^)	(mg/cm^3^)	(mg/cm^3^)
Normal (*n* = 536)	134.78 ± 24.41	168.49 ± 23.31	123.19 ± 24.25	61.59 ± 10.74	80.52 ± 10.98	948.06 ± 12.42
Osteopenia (*n* = 442)	91.12 ± 14.89	127.92 ± 15.38	82.01 ± 14.87	42.85 ± 7.09	61.34 ± 7.07	946.13 ± 11.57
Osteoporosis (*n* = 393)	59.47 ± 15.65	97.72 ± 16.35	52.03 ± 15.99	28.63 ± 7.61	46.63 ± 7.83	943.91 ± 13.38
Statistical value	956.396	899.58	910.072	901.6	908.129	25.979
*p*	0.001	0.001	0.001	0.001	0.001	0.001

**Table 4 diagnostics-13-01751-t004:** ROC curve analysis of DECT base material pairs for the diagnosis of osteopenia.

	HAP (Water)	HAP (Fat)	HAP (Blood)	Ca (Water)	Ca (Fat)	Fat (HAP)
	(mg/cm^3^)	(mg/cm^3^)	(mg/cm^3^)	(mg/cm^3^)	(mg/cm^3^)	(mg/cm^3^)
AUC	0.953	0.930	0.934	0.930	0.932	0.556
95% CI	0.938–0.965	0.913–0.945	0.917–0.949	0.913–0.945	0.915–0.947	0.525–0.587
*p* value	<0.0001	<0.0001	<0.0001	<0.0001	<0.0001	<0.0001
Youden index J	0.7579	0.7045	0.7148	0.7086	0.7182	0.1103
Criterion	≤107.4	≤144.3	≤97.35	≤50.04	≤68.35	≤949
Sensitivity	86.88	84.62	83.94	83.48	83.94	60.86
Specificity	88.91	85.84	87.54	87.37	87.88	50.17

**Table 5 diagnostics-13-01751-t005:** ROC curve analysis of DECT base material pairs for the diagnosis of osteoporosis.

	HAP (Water)	HAP (Fat)	HAP (Blood)	Ca (Water)	Ca (Fat)	Fat (HAP)
	(mg/cm^3^)	(mg/cm^3^)	(mg/cm^3^)	(mg/cm^3^)	(mg/cm^3^)	(mg/cm^3^)
AUC	0.999	0.996	0.997	0.997	0.997	0.594
95% CI	0.995–1.000	0.990–0.999	0.991–0.999	0.991–0.999	0.991–0.999	0.563–0.625
*p* value	<0.0001	<0.0001	<0.0001	<0.0001	<0.0001	<0.0001
Youden index J	0.9876	0.9001	0.8758	0.9514	0.9498	0.1732
Criterion	≤89.62	≤122.3	≤86.91	≤44.85	≤63.16	≤945.9
Sensitivity	99.24	95.67	99.24	99.24	99.24	57.25
Specificity	99.53	94.33	88.35	95.91	95.75	60.07

## Data Availability

The datasets used and analyzed in the current study are available from the corresponding author upon reasonable request.

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
