# Peer review of "Diagnostic Accuracy of Dual-Energy CT Material Decomposition Technique for Assessing Bone Status Compared with Quantitative Computed Tomography"

_diagnostics, 2023, doi:10.3390/diagnostics13101751_

Round 1

Reviewer 1 Report

1. The human ethical approval code and the guideline used should be included in the materials and methods.

2. Materials and methods: It is recommended to provide some definition/explanation for “hydroxyapatite(water), hydroxyapatite(fat), hydroxyapatite(blood), calcium(water), and calcium(fat) [DHAP(water), DHAP(fat), DHAP(blood), DCa(water), and DCa(fat)]”. Which area do these parameters measure and what do these parameters indicate? These will provide better understanding to the readers.

3. Figure 1: It is recommended to include the initial number of vertebral bodies (for T11, T12 and L1) before exclusion since authors have indicated the final number of vertebral bodies.

4. Figure 2: The labelling of the bone is needed. Which picture represent which vertebral bodies (T11, T12 and L1)?

5. Section 3.1: The demographic data of the population should be presented in table.

6. The authors mention that “There is still no universal agreement on the threshold value in DECT for diagnosing osteoporosis and osteopenia” as the research gap to be filled in the introduction. Does this research gap achieve in this study?

7. In most cases, diagnosis of osteoporosis using spine will be performed at lumbar. This study used only L1 for the lumbar region. How does this differ in clinical setting?

8. A recent study has been conducted which is related to DECT (10.1007/s00330-021-08323-9). It is recommended to include/cite this study and indicate what are the differences between current and previous studies

Author Response

Dear Editors and Reviewers,

On behalf of all the contributing authors, I sincerely thank you for giving us an opportunity to revise our manuscript. I would like to express our appreciation for your constructive comments on our manuscript entitled “Diagnostic accuracy of dual-energy CT material decomposition technique for assessing bone status compared with quantitative computed tomography” (diagnostics-2337267). These comments are valuable and help us to improve our article. I have considered the comments very carefully and have revised the paper accordingly. All the changes/additions were highlighted within the manuscript file by using the red font color. The summary of changes and an item-by-item response to the reviewers’ comments are listed below this letter. Finally, we appreciate your warm work earnestly and hope the revised manuscript could be acceptable for you. 

Best wishes,

Xu Wang

The Reviewer’s comment:

  1. The human ethical approval code and the guideline used should be included in the materials and methods.
  2. Materials and methods: It is recommended to provide some definition/explanation for “hydroxyapatite(water), hydroxyapatite(fat), hydroxyapatite(blood), calcium(water), and calcium(fat) [DHAP(water), DHAP(fat), DHAP(blood), DCa(water), and DCa(fat)]”. Which area do these parameters measure and what do these parameters indicate? These will provide better understanding to the readers.
  3. Figure 1: It is recommended to include the initial number of vertebral bodies (for T11, T12 and L1) before exclusion since authors have indicated the final number of vertebral bodies.
  4. Figure 2: The labelling of the bone is needed. Which picture represent which vertebral bodies (T11, T12 and L1)?
  5. Section 3.1: The demographic data of the population should be presented in table.
  6. The authors mention that “There is still no universal agreement on the threshold value in DECT for diagnosing osteoporosis and osteopenia” as the research gap to be filled in the introduction. Does this research gap achieve in this study?
  7. In most cases, diagnosis of osteoporosis using spine will be performed at lumbar. This study used only L1 for the lumbar region. How does this differ in clinical setting?
  8. A recent study has been conducted which is related to DECT (10.1007/s00330-021-08323-9). It is recommended to include/cite this study and indicate what are the differences between current and previous studies

The authors’ answer: We sincerely appreciate the valuable comments. We have revised my manuscript according to your suggestions.

  1. We have added a statement of human ethics Approval Guidelines and corresponding ethical document batch numbers to the materials method.
  2. As you suggested, in Materials and Methods (2.1. Population and study design), the basic materials are explained, and the specific method of parameter measurement is explained in detail for readers to better understand.
  3. The initial number of vertebrae is given in Figure 1.
  4. The vertebrae in Figure 2 are marked in detail.
  5. Thank you very much for the advice. We try to present the demographic data of this study in tabular form.
  6. Our study preliminically realized the threshold value for the diagnosis of osteoporosis and osteopenia by DECT. Currently, there is no general consensus, which is a research gap to be filled. The significance of this study is reiterated in the discussion, corresponding to the preface.
  7. We deeply appreciate the thoughtful advice about the lumbar region. Our consideration for the lumbar spine up to L1 is that thoracic and abdominal CT examinations generally include the L1 vertebra, which is considered to be the best target for screening for opportunistic osteoporosis in routine CT examinations. Compared with other vertebrae, the L1 vertebra is easily recognized and less prone to degenerative changes. Therefore, the measurement of L1 bone density has important clinical significance.
  8. As you suggested, we added the reference that you recommend to the discussion about some difference between current and previous studies.

We highly appreciate the valuable comments of the Reviewer, which are very helpful not only in improving current manuscript but also in our future works.

Reviewer 2 Report

Title

Ok

Abstract

fine

INTRODUCTION

Overall is well written. I would try to summarize the description of DXA and QCT. First one is no part of the study, while the other one is used as reference. Conversely I would better describe the role of DECT and its importance in diagnosis of vertebral compression fractures. The number of DECT are increasing in clinical practice and its role is becoming more and more important. In this scenario DECT may represent a one stop one shop exam, allowing diagnosis of BME but also the measurement of bone density.

I suggest to include this idea to corroborate the rationale of the study and to include some additional references describing the role of DECT in depiction of vertebral fractures and in general the presence of bone marrow edema in traumatic and non traumatic settings. Bone marrow edema is the typical response of osteoporotic bone in vertebral compression fractures, but also in transient bone marrow lesions or in stress or insufficiency fractures. These topics may be useful to give some color to discussion as well.

METHODS

-        Clarify if abdominal CT were with or without contrast material

-        Specify if patients were consecutive and if they were referred from specific clinicians

-        Describe number of Readers and experience of each reader

-        In statistical analysis please clarify regarding numbers of readers and type of reading sessions

-        Also clarify if all the 3 vertebrae were assessed or only the results of 1 vertebra

-         

RESULTS

Well described

DISCUSSION

I suggest to underline again the possibility to use DECT for diagnosing vertebral fractures. From this underline that is possible to measure density of vertebral body when a patient comes for diagnosis of fracture.

There are some other recent paper describing DECT in diagnosis of osteoporosis, using for example DEXA as reference; I suggest to briefly compare your results with those of these studies.

Before limitations include a paragraph to describe the potential changes in scientific and clinical fields coming from the results of this paper.

Author Response

Dear Editors and Reviewers,

On behalf of all the contributing authors, I sincerely thank you for giving us an opportunity to revise our manuscript. I would like to express our appreciation for your constructive comments on our manuscript entitled “Diagnostic accuracy of dual-energy CT material decomposition technique for assessing bone status compared with quantitative computed tomography” (diagnostics-2337267). These comments are valuable and help us to improve our article. I have considered the comments very carefully and have revised the paper accordingly. All the changes/additions were highlighted within the manuscript file by using the red font color. The summary of changes and an item-by-item response to the reviewers’ comments are listed below this letter. Finally, we appreciate your warm work earnestly and hope the revised manuscript could be acceptable for you. 

Best wishes,

Xu Wang

The Reviewer’s comment:

INTRODUCTION

Overall is well written. I would try to summarize the description of DXA and QCT. First one is no part of the study, while the other one is used as reference. Conversely I would better describe the role of DECT and its importance in diagnosis of vertebral compression fractures. The number of DECT are increasing in clinical practice and its role is becoming more and more important. In this scenario DECT may represent a one stop one shop exam, allowing diagnosis of BME but also the measurement of bone density.

I suggest to include this idea to corroborate the rationale of the study and to include some additional references describing the role of DECT in depiction of vertebral fractures and in general the presence of bone marrow edema in traumatic and non traumatic settings. Bone marrow edema is the typical response of osteoporotic bone in vertebral compression fractures, but also in transient bone marrow lesions or in stress or insufficiency fractures. These topics may be useful to give some color to discussion as well.

METHODS

Clarify if abdominal CT were with or without contrast material

Specify if patients were consecutive and if they were referred from specific clinicians

Describe number of Readers and experience of each reader

In statistical analysis please clarify regarding numbers of readers and type of reading sessions

Also clarify if all the 3 vertebrae were assessed or only the results of 1 vertebra

DISCUSSION

I suggest to underline again the possibility to use DECT for diagnosing vertebral fractures. From this underline that is possible to measure density of vertebral body when a patient comes for diagnosis of fracture.

There are some other recent paper describing DECT in diagnosis of osteoporosis, using for example DEXA as reference; I suggest to briefly compare your results with those of these studies.

Before limitations include a paragraph to describe the potential changes in scientific and clinical fields coming from the results of this paper.

The authors’ answer: We feel great thanks for your professional review work on our article. As you mentioned above, there are several problems that need to be addressed. According to your valuable suggestion, we have made corrections to our manuscript.

-I quite agree with your comments and suggestions on the introduction. I tried to summarize the description of DXA and QCT, and expanded the clinical application value of DECT, especially in the aspects of vertebral compression fracture, bone marrow edema, tumor infiltration and other advantages. In addition, DECT is expected to provide a one-stop examination to diagnose vertebral compression fractures and obtain additional vertebral bone density information.

-Thank you very much for your suggestion. Maybe it is not clearly expressed in our manuscript. In our study, patients with chest and abdomen plain scan were not treated with contrast media.

-Secondly, patients were continuously collected and no clinicians were referred.

-Meanwhile, the number and qualifications of physicians measured by the data were described in detail. The QCT data and DECT data were evaluated in random order by two radiologists with more than 10 years of experience in musculoskeletal radiology, who were independent and blinded to each other’s results. The data were measured twice by each radiologist.

-Statistical analysis was performed independently by a physician with 3 years of experience in medical statistics.

-Three vertebrae of patients were included in our study, but single vertebrae was used as the study object for data analysis. As explained in the discussion in this paper: The purpose of our study was to compare the differences in BMD measurements be-tween QCT-based BMD and DECT-based density measurements. Moreover, during QCT BMD measurements, it was found that determining bone status by the mean BMD values of equal to or greater than two vertebrae would lead to the underestimation of the bone status in patients with low bone mass, which is the reason why a single vertebra was the analysis object in our study.

The above question has been described in more detail in the Materials and Methods section of this paper.

-As you suggested, In the discussion, the use of DECT to obtain vertebral bone density information, early detection and diagnosis of vertebral fractures was emphasized again. At the same time, we added the related research discussion of DXA, and compared with the results of this study. The positive clinical implications of the results of this study are supplemented before limitations.

We highly appreciate the valuable comments of the Reviewer, which are very helpful not only in improving current manuscript but also in our future works.

To editors:

Thank you for your kind letter and your careful work regarding our manuscript. We have revised the format questions by referring to the journal manuscript sample. In addition, I found that the author's name was reversed in the manuscript, and the corresponding author did not correspond to the manuscript submission system, so I adjusted it on my own.
